# Validity of the models predicting 10-year risk of cardiovascular diseases in Asia: A systematic review and prediction model meta-analysis

Mahin Nomali[1], Davood Khalili[2], Mehdi Yaseri [1], Mohammad Ali Mansournia[1], Aryan Ayati[3], Hossein Navid[3], Saharnaz Nedjat [1]*

1 Department of Epidemiology and Biostatistics, School of Public Health, Tehran University of Medical Sciences, Tehran, Iran, 2 Research Institute for Endocrine Sciences, Prevention of Metabolic Disorders Research Center, Shahid Beheshti University of Medical Sciences, Tehran, Iran, 3 Tehran Heart Center, Cardiovascular Diseases Research Institute, Tehran University of Medical Sciences, Tehran, Iran

* nejatsan@tums.ac.ir

**Data Availability Statement:** All relevant data are within the paper and its Supporting Information files.

## Abstract

We aimed to review the validity of existing prediction models for cardiovascular diseases (CVDs) in Asia. In this systematic review and meta-analysis, we included studies that validated prediction models for CVD risk in the general population in Asia. Various databases, including PubMed, Web of Science conference proceedings citation index, Scopus, Global Index Medicus of the World Health Organization (WHO), and Open Access Thesis and Dissertations (OATD), were searched up to November 2022. Additional studies were identified through reference lists and related reviews. The risk of bias was assessed using the PRO-BAST prediction model risk of bias assessment tool. Meta-analyses were performed using the random effects model, focusing on the C-statistic as a discrimination index and the observed-to-expected ratio (OE) as a calibration index. Out of 1315 initial records, 16 studies were included, with 21 external validations of six models in Asia. The validated models consisted of Framingham models, pooled cohort equations (PCEs), SCORE, Globorisk, and WHO models, combined with the results of the first four models. The pooled C-statistic for men ranged from 0.72 (95% CI 0.70 to 0.75; PCEs) to 0.76 (95% CI 0.74 to 0.78; Framingham general CVD). In women, it varied from 0.74 (95% CI 0.22 to 0.97; SCORE) to 0.79 (95% CI 0.74 to 0.83; Framingham general CVD). The pooled OE ratio for men ranged from 0.21 (95% CI 0.018 to 2.49; Framingham CHD) to 1.11 (95%CI 0.65 to 1.89; PCEs). In women, it varied from 0.28 (95%CI 0.33 to 2.33; Framingham CHD) to 1.81 (95% CI 0.90 to 3.64; PCEs). The Framingham, PCEs, and SCORE models exhibited acceptable discrimination but poor calibration in predicting the 10-year risk of CVDs in Asia. Recalibration and updates are necessary before implementing these models in the region.

**Funding:** The author(s) received no specific funding for this work.

**Competing interests:** The authors have declared that no competing interests exist.

## Introduction

Cardiovascular diseases (CVDs) are the leading cause of worldwide disease burden. The number of total CVD cases has nearly doubled from 271 million in 1990 to 523 million in 2019, and the number of CVD deaths increased from 12.1 million in 1990 to 18.6 million in 2019 [1]. In Asia, CVD is the major cause of death [2], in which CVD deaths surged from 5.6 million to 10.8 million between 1990 and 2019 [3].

It is expected that the number of people affected by CVD will significantly rise due to factors such as population growth and aging, particularly in Northern Africa and Western Asia, Central and Southern Asia, Latin America and the Caribbean, and Eastern and Southeastern Asia, where the percentage of elderly individuals is anticipated to double by 2050 [1].

Since Asian countries have become more Westernized, they have been eating more fat, which has led to an increase in serum total cholesterol, and may have contributed to a rise in coronary heart disease in this region [4].

The burden of CVD varies regionally and nationally due to differences in the incidence of risk factors as well as the availability of medical care [5]. Some risk factors cannot be removed entirely to prevent CVDs. Therefore, prevention may be a valuable way to manage CVDs in the general population [6], and using risk prediction models to target the population is ongoing [7]. It is also recommended by the American College of Cardiology/ American Heart Association (ACC/AHA) [8], the European Society of Cardiology (ESC) [9], and HEARTS [10] guidelines.

Various models have been developed in Western countries to estimate the risk of CVDs [11–19]. However, differences in risk profiles and access to healthcare facilities may prevent being applied to Asia without any validation and performance assessment [20,21]. In addition, a systematic review by Damen et al. in 2016 indicated that only 19% of developed models for CVDs have been externally validated by independent researchers [22].

Although previous systematic reviews and meta-analyses evaluated the performance of the externally validated Framingham models and pooled cohort equations (PCEs) for the general population in 2019 [23] and the general Chinese population in 2022 [24], the performance of existing prediction models are not adequately assessed in Asia [21,25–27]. For the first time, we aimed to review the performance of existing prediction models for predicting the 10-year risk of CVDs in Asia through a systematic review and meta-analysis. Our findings demonstrate a complete picture of the CVD model's performance in Asia, which helps researchers to choose the best model for designing prediction model studies, and clinicians and policymakers to choose the best one to apply in the prevention programs.

## Materials and methods

This was a systematic review and prediction model meta-analysis, which was approved by the Research Ethics Committee (REC) of the School of Public Health & Allied Medical Sciences at TUMS on March 12, 2022 (approval ID: IR.TUMS.SPH.REC.1400.353).

### Eligibility criteria

In this review, studies that externally validated existing prediction models to estimate the 10-year risk of CVDs for both men and women in the general population in Asia were included. The PICOTS components were defined as follows:

- Participants (P): General population in Asia

- Intervention (I) and comparators (C): independent investigators externally validate CVD prediction models in Asia.

- Outcome (O): Outcome for which the original CVD models were developed (i.e., risk of CVD)

- Timing/prediction horizon (T): 10 years

- Setting (S): Primary healthcare

In addition, we excluded review articles, studies that developed new models instead of validating the existing prediction models, and studies that validated models for vascular diseases or stroke or validated exclusively in patients or an area other than Asia or with inadequate data regarding model performance measures. We also excluded studies that performed external validation as a phase of model development in Western countries. It should be noted that we did not consider any language or time limitations.

## Information sources and search strategy

Our data sources were databases, including PubMed, Web of Science conference proceedings citation index, Scopus, Global Index Medicus of the World Health Organization (WHO), and Open Access Thesis and Dissertations (OATD).

We searched in PubMed through the following search strategy: (("Cardiovascular Diseases"[Mesh]) AND ("risk chart" OR "risk score" OR "risk equation" OR "risk algorithm" OR "risk prediction" OR "risk assessment") AND (validation OR calibration) AND (Asia OR "Middle east")) from inception until November 28, 2022. The search strategy for other databases was modified, as shown in **S1 Table**. We also checked the reference list of included studies and related reviews to identify additional studies.

## Selection process

Two authors (MN and DK) independently screened the retrieved records based on title and abstract. After that, we evaluated the full text of the papers and selected those that met the mentioned eligibility criteria. Any disagreement was resolved by discussion with the third person (SN).

## Data items and data extraction process

We extracted the following variables from the included models: the first author's name, publication year, study country, WHO region using www.who.int/countries, prediction model, population (i.e., the data source used for external validation), predictors, follow-up time, study outcome and outcome definition, sample size (i.e., total and in men and women, separately), number of events, statistical modeling method and analysis (i.e., handling missing data), and model performance indices, including C-statistic for discrimination and observed-to-expected (OE) ratio for calibration.

To calculate the observed-to-expected ratio, we extracted the total sample size and the total number of observed (O) and expected (E) events across gender [28]. Otherwise, we used WebPlotDigitizer (version 4.6) to get the probability for the decile of risks [29]. For model discrimination, we obtained the C-statistic and the 95% confidence interval (CI) or standard error (SE) for men and women separately. For articles that validated multiple models in one study, we did separate data extractions for each model.

One reviewer (MN) screened the full text of included studies for data extraction, and relevant data were extracted. Another reviewer (MY) checked the extracted items. A third reviewer (DK) also reached a consensus regarding any disagreements.

## Risk of bias assessment

To assess the risk of bias in included studies, we used the prediction model risk of bias assessment tool (PROBAST) [30]. It contained 20 signaling questions across four domains of participants, predictors, outcome, and analysis, which were answered as yes, no, and no information. Finally, the results were summarized into three groups: low, high, or unclear risk of bias. The methodological quality of included studies was assessed by two independent reviewers (MN, MM). We resolved disagreements by discussing and consulting with a third person (SN) to reach a consensus.

## Study measures

This study's measures were C-statistic, which indicates model discrimination, and OE ratio, which indicates model calibration. Discrimination was the ability of a model to separate high-risk individuals from low-risk individuals and was typically reported as a measure of concordance, the c-statistic, which varies between 0.5 and 1.0 [31]. The C-statistic of 0.5–0.7, 0.70–80, 0.80–0.90, and $\geq$0.90 indicated poor, acceptable, excellent, and outstanding discriminative ability, respectively [32]. Calibration was the agreement between observed and expected outcomes. The OE ratio less than 1 indicated overestimation, and the OE greater than 1 indicated underestimation of the risk [33].

## Statistical analysis

The OE ratio and its standard error were calculated using the equations provided in Debray et al. study [26] to estimate the total O: E ratio from other information in a preliminary study.

Meta-analyses of the OE ratio and the C-statistic for CVD models were performed by random effects model with restricted maximum likelihood estimation [34] and 95% confidence intervals (CI) to quantify the uncertainty of the pooled performance estimates.

As previously recommended for the meta-analysis of prediction models [34], we also reported 95% prediction intervals (PI) to depict the between-study heterogeneity. The PI sets limits on the expected performance of future model validation, which will be comparable to the studies included in the meta-analysis.

Because of methodological heterogeneity between various prediction models, we did not provide all study combinations, and we provided the meta-analysis stratified by model and gender. In addition, some studies have reported a 5-year risk of CVDs due to short follow-up, despite using CVD models originally developed to predict a 10-year risk. In this case, we extrapolated observed and expected event probabilities to 10 years using the formula based on the Poisson distribution [23,28,34].

We also did sensitivity analysis by excluding studies that reported 5-year risk evaluating their effect on the overall model performance. In addition, despite the recommendation of previous studies to perform the random effects-model meta-analysis for prediction model studies [28,34], we also quantitatively synthesized the performance measures using the fixed-effect model for the Framingham CHD and SCORE risk models because of being validated twice [35]. We used R version 4.2.2, the metamisc package, and the valmeta function to perform the meta-analyses. For a graphical depiction of the PROBAST risk of bias assessment, we used STATA/IC version 14.2 (Stata Corp LP College Station, TX, USA) and the QUADAS module.

# Results

## Study selection and characteristics

From 1315 records identified through searching in databases and the reference list for included studies and related reviews [21–27,36], a total of 50 full texts were reviewed to assess whether they met the eligibility criteria, and ultimately, 16 studies were included in the review. **Fig 1** illustrates the process of searching, screening, and selecting records.

Five studies [38–42] simultaneously validated two risk models. Therefore, 21 prediction models were validated by 16 included studies. Fifteen studies were published in English, and only one was written in Chinese [43], which were considered in the review regardless of language.

**Table 1** shows the characteristics of validated risk models in Asia. According to **Table 1**, Framingham coronary heart disease (CHD) [44], Framingham general CVD [45], PCEs [46], SCORE [47], Globorisk [19], and WHO [48] models were risk scores externally validated in Asia. Out of 21 validated models, 15 models (71%) were validated in the Western Pacific region of the WHO (i.e., in China, Japan, and Korea), and six models (29%) were validated in the Eastern Mediterranean region (i.e., in Iran). There were external validations of the Framingham general CVD [45], PCEs [46], and SCORE [47] risk models in both the Eastern Mediterranean and Western Pacific regions. In contrast, the Framingham CHD [44] was only validated in the Western Pacific region. The Globorisk [19] and WHO [48] risk models were validated for the Eastern Mediterranean and Western Pacific regions, respectively (**Table 1**).

## Risk of bias assessment

**Fig 2** shows a summary of the risk of bias analysis by PROBAST. Two domains of participant selection and outcome had a low risk of bias. In only one study, the predictors were unclear, and others were at low risk of bias for predictors. In terms of the analysis domain, more than 50% of the studies were judged to have a high risk of bias because they did not adequately handle missing data and used a complete case approach instead of multiple imputations. Thus, most studies were scored high risk of bias (**Fig 2**). Additional detailed information has been provided in **S2 Table.**

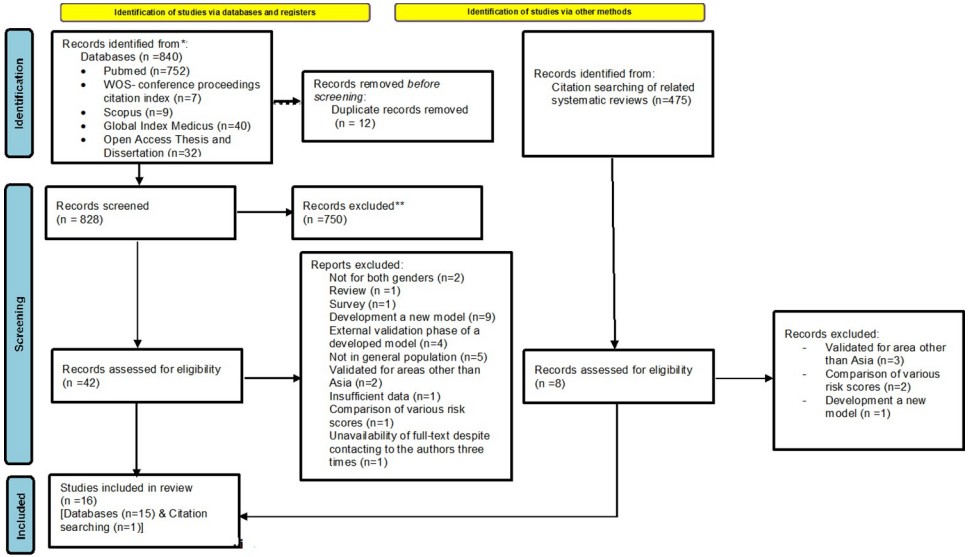

**Fig 1. PRISMA 2020 [37] flow diagram of the process of searching, screening and selecting records.**

**Table 1. The characteristics of validated risk models in Asia.**

| Reference | Predictors | Country (WHO region) | Data source & follow-up time | Study Population | Outcome (Definition) | Sample Size (Outcome) | Modeling method | Model Predictive Performance | | |
|---|---|---|---|---|---|---|---|---|---|---|
| | | | | | | | | Discrimination | Calibration | Clinical usefulness |
| **RISK SCORE: Framingham CHD [44]** | | | | | | | | | | |
| Liu et al. 2004 [49] | Age, BP, smoking, diabetes, TC, HDL-C | China (Western Pacific Region) | Chinese Multi-provincial Cohort Study (CMCS) Follow-up time: 10 years | Aged 35 to 64 years, without clinical history of MI/ angina pectoris at baseline | 10-year risk of CHD (i.e. death & MI) | 30121 individuals, 16065 men & 14056 women (816 events) | CoxPH | C-statistic (95% CI): In men: 0.705 (0.665,0.746) In women: 0.742 (0.686, 0.798) | Calibration plot The H-L $\chi$ 2: In men 645.9 (P<0.001) In women 147.6 (P<0.001) | -- |
| Jee et al. 2014 [50] | Age, BP, smoking, diabetes, TC | Korea (Western Pacific Region) | Korean Heart Study (KHS), a prospective cohort study Follow-up time (median): 11.6 years | Aged 30 to 74 years Without CHD at baseline | 10-year risk of CHD (i.e. acute MI, sudden death & other coronary deaths) | 268315 individuals, 164005 men & 104310 women (2596 events) | CoxPH | C-statistic (95% CI): In men: 0.756 (0.745, 0.766) In women: 0.809 (0.789, 0.829) | Calibration plot | -- |
| **RISK SCORE: Framingham general CVD [45]** | | | | | | | | | | |
| Bozorgmanesh et al. 2011 [51] | Age, SBP, antihypertensive treatment, TC, HDL-C, smoking, diabetes mellitus | Iran (Eastern Mediterranean Region) | TLGS, a population-based cohort study Follow-up time (median): 8.6 years | Aged ≥ 30 years | **5-year risk of CVD** (i.e. any CHD events, stroke, or cerebrovascular death) | 3838 individuals, 1655 men & 2183 women (283 events) | CoxPH | C-statistic (95% CI): In men: 0.778 (0.744, 0.812) In women: 0.839 (0.813,0.864) | Calibration plot Nam-D'Agostino $\chi$ 2: In men 20.2 (P = 0.014) In women 5.0 (P = 0.831) | -- |
| Khalili et. al. 2012 [52] | Age, HTN medication, diabetes medication, smoking, SBP, TC, HDL-C, FPG | Iran (Eastern Mediterranean Region) | TLGS, a population-based prospective cohort study Follow-up time (median): 9.3 years | Aged 30 to 74 years | 10-year risk of CVD (i.e. any CHD, CHD death, HF and cerebrovascular events) | 6224 individuals; 2640 men & 3584 women (515 events) | CoxPH | C-statistics (Jackknife SE): In men: 0.785 (0.012) In women: 0.832 (0.012) | Calibration plot The H-L $\chi$ 2: In men: 24.2 In women: 27.8 | DCA |
| Lee et.al. 2015 [41] | Age, current smoker, BMI, waist circumference, diabetes, fasting glucose, SBP, DBP, treated HTN, TC, TG, HDL-C, LDL-C | Hong Kong, China (Western Pacific Region) | Hong Kong Cardiovascular Risk Factor Prevalence Study (CRISPS) population-based cohort study Follow-up time: 10 years | Aged 30 to 74 years | 10-year risk of CVDs (i.e. fatal or nonfatal MI, coronary insufficiency, angina, stroke, TIA, PVD, HF, and CHD or stroke-related mortality) | 1668 individuals; 771 men & 917 women (138 events) | Framingham CV risk equation | C-statistic (95% CI): In men: 0.773 (0.742, 0.802) In women: 0.788 (0.724,0.852) | Calibration plot The H-L $\chi$ 2: In men: 20.1 In women: 12.1 | -- |

*(Continued)*

**Table 1.** (Continued)

| Reference | Predictors | Country (WHO region) | Data source & follow-up time | Study Population | Outcome (Definition) | Sample Size (Outcome) | Modeling method | Model Predictive Performance | | |
|---|---|---|---|---|---|---|---|---|---|---|
| | | | | | | | | Discrimination | Calibration | Clinical usefulness |
| Sepanlou et. al 2015 [53] | Age, BMI, SBP, treatment of hypertension, current smoking, diabetes | Iran (Eastern Mediterranean Region) | Golestan Cohort in North-East of Iran Follow-up time (median): 7.1 years | Aged 40 to 75 years from January 2004 to June 2008 Living in Gonbad city & 326 villages in Golestan province (northeast of Iran) | 10-year risk of fatal CVDs (i.e. deaths due to CVD, including IHD, cerebrovascular diseases, & all other vascular diseases) | 46674 individuals; 19820 men & 26854 women (1438 events) | CoxPH | C-statistic (95% CI): In men: 0.763 (0.747,0.779) In women: 0.772 (0.753,0.791) | Calibration plot | --- |
| **RISK SCORE: Framingham general CVD [45]** | | | | | | | | | | |
| Bae et al. 2020 [40] | Age, SBP, TC, HDL-C, BP treatment, diabetes, current smoker | Korea (Western Pacific Region) | Korean Genome and Epidemiology Study (KOGES), a community based cohort study Follow-up time [mean (SD)]: 8.4 (2.7) years | Aged 40 to 69 years Without a history of ASCVD at baseline No missing data to calculate ASCVD risk Without lost to follow-up after baseline study | 10-year risk of ASCVD (i.e. first nonfatal MI, UA, stable angina pectoris, nonfatal ischemic stroke, TIA, and death from ASCVD) | 7932 individuals; 3778 men & 4154 women (598 events) | CoxPH | C-statistic (95% CI) In men: 0.730 (0.715, 0.744) In women: 0.726 (0.712, 0.739) | Calibration plot The H-L $\chi$ 2: In men 177.71 (P<0.001) In women 24.70 (P = 0.002) | --- |
| Jiang et al 2020 [39] | Age, current smoker, treated SBP, TC, HDL-C, diabetes | China (Western Pacific Region) | Prospective cohort Follow-up time (median): 7.05 years | Aged 40 to 74 years Without a history of CVD at baseline | **5-year risk of ASCVD** (i.e. nonfatal MI or CHD death, or fatal or nonfatal stroke) | 3347 individuals; 1508 men & 1839 women (286 events) | Framingham risk equations | C-statistic (95% CI) In men: 0.740 (0.703, 0.777) In women: 0.761 (0.728, 0.794) | Calibration plot Nam-D'agostino calibration $\chi$ 2: In men: 13.58 In women: 48.13 EO ratio: In men: 1.06 In women: 0.49 | DCA |
| **RISK SCORE: PCEs [46]** | | | | | | | | | | |

(Continued)

**Table 1.** (Continued)

| Reference | Risk score & Predictors | Country (WHO region) | Data source & follow-up time | Study Population | Outcome (Definition) | Sample Size (Outcome) | Modeling method | Model Predictive Performance | | |
|---|---|---|---|---|---|---|---|---|---|---|
| | | | | | | | | Discrimination | Calibration | Clinical usefulness |
| Lee et.al. 2015 [41] | Age, current smoker, BMI, waist circumference, diabetes, fasting glucose, SBP, DBP, treated HTN, TC, TG, HDL-C, LDL-C | Hong Kong, China (Western Pacific Region) | Hong Kong Cardiovascular Risk Factor Prevalence Study (CRISPS) population-based cohort study Follow-up time: 10 years | Aged 40 to 79 years | 10-year risk of ASCVD (i.e. all-fatal or nonfatal MI, stroke, & CHD or fatal stroke) | 1476 individuals; 679 men & 797 women (122 events) | PCEs recommended by ACC/AHA | C-statistic (95% CI): In men: 0.714 (0.657, 0.770) In women: 0.765 (0.690, 0.840) | Calibration plot The H-L $\chi$ 2: In men: 24.1 In women: 10.1 | --- |
| Jung et al. 2015 [54] | Age, BP, TC and HDL-C, diabetes, smoking status | Korea (Western Pacific Region) | Korean Heart Study (KHS), a cohort from 1996 to 2001 **Follow-up time (mean):** 12.8 years | Aged 40 to 79 years Free from ASCVD Without consumption of lipid-lowering medications Without missing Values for predictors at baseline | 10-year risk of ASCVD (i.e. CHD death or fatal stroke or nonfatal MI or stroke) | 192605 individuals; 114622 men & 77983 women (12327 events) | CoxPH | C-statistic (95% CI): In men: 0.727 (0.721, 0.734) In women: 0.738 (0.729, 0.746) | Calibration plot The H-L $\chi$ 2: In men 1364.26 (P < 0.001) In women 683.12 (P < 0.001) | --- |
| Khalili et al 2015 [55] | Age, treated or untreated SBP, TC, HDL-C, current smoking status, and history of diabetes | Iran (Eastern Mediterranean Region) | TLGS, a population-based prospective cohort study Follow-up time (median): 10.1 years | Aged 40 to 75 years | 10-year risk of ASCVD (i.e. CHD and CHD death, cerebrovascular events and cerebrovascular death) | 5102 individuals; 2353 men & 2749 women (726 events) | Pooled Risk Equations recommended by ACC/AHA | C-statistics (95% CI) In men: 0.74 (0.71, 0.78) In women: 0.82 (0.78,0.86) | Calibration plot The H-L $\chi$ 2: In men 12.9 In women 14.7 | DCA |
| Tang et al. 2017 [43] | Age, current smoker, waist circumference, SBP, DBP, antihypertensive treatment, TC, HDL-C, diabetes, family history of ASCVD | China (Western Pacific Region) | Contemporary rural Northern Chinese population Follow-up time: 5.82 years | Aged 40 to 79 years Rural adults without clinical ASCVD | 5-year risk of ASCVD (nonfatal MI, CHD death, nonfatal or fatal stroke) | 6489 individuals; 2217 men & 4272 women (955 events) | PCE recommended by ACC/AHA | C-statistics (95% CI) In men: 0.702 (0.675, 0.730) In women: 0.714 (0.695, 0.773) | Calibration plot The H-L $\chi$ 2: **In men:** 192 (P<0.001) **In women:** 181.2 (P<0.001) E/O ratio: In men: 1.673 In women: 1.531 | --- |

**RISK SCORE: PCEs**

(*Continued*)

**Table 1.** (Continued)

| Study | Predictors | Country/Region | Cohort | Population/Criteria | Outcome | Sample size | Method | C-statistic (95% CI) | Calibration | Other |
|---|---|---|---|---|---|---|---|---|---|---|
| Tang et al. 2019 [56] | Age, current smoker, waist circumference, SBP, DBP, antihypertensive treatment, TC, HDL-C, diabetes, family history of ASCVD | China (Western Pacific Region) | Fangshan Cohort Study (FCS), a prospective population-based cohort in rural Beijing in North China Follow-up time: Median of 6.44 years | Aged 40 to 79 years without a history of CHD, stroke, HF, or AF at baseline No missing date on components of the PCE models. | 5-year risk of ASCVD (i.e. nonfatal or fatal stroke, nonfatal MI, and CHD death) | 11169 individuals; 3578 men & 7591 women (1921 events) | CoxPH | In men: 0.675 (0.649,0.701) In women: 0.714 (0.697,0.731) | Calibration plot The H-L χ 2: In men 2218.7 (P <0 .001) In women 6432.1 (P < 0.001) E/O ratio: In men: 0.238 In women: 0.118 | -- |
| Bae et al. 2020 [40] | Age, SBP, TC, HDL-C, BP treatment, diabetes, current smoker | Korea (Western Pacific Region) | Korean Genome and Epidemiology Study (KOGES), a community based cohort study Follow-up time [mean (SD)]: 8.4 (2.7) years | Aged 40 to 69 years Without a history of ASCVD at baseline No missing data to calculate ASCVD risk Without lost to follow-up after baseline study | 10-year risk of ASCVD (i.e. first nonfatal MI, UA, stable angina pectoris, nonfatal ischemic stroke, TIA, and death from ASCVD) | 7932 individuals; 3778 men & 4154 women (598 events) | CoxPH | In men: 0.731 (0.717, 0.745) In women: 0.726 (0.712, 0.739) | Calibration plot The H-L χ 2: In men 10.59 (P = 0.226) In women 258.62 (P<0.001) | -- |
| Jiang et al 2020 [39] | Age, current smoker, treated or untreated SBP, TC, HDL-C, diabetes | China (Western Pacific Region) | Prospective cohort Follow-up time (median): 7.05 years | Aged 40 to 74 years Without CVD at baseline | 5-year risk of ASCVD (i.e. nonfatal MI or CHD death, or fatal or nonfatal stroke) | 3347 individuals; 1508 men & 1839 women (286 events) | Pooled cohort equations | In men: 0.727 (0.689,0.766) In women: 0.738 (0.703,0.773) | Calibration plot Nam-D'agostino calibration χ 2: In men: 39.86 In women: 86.26 E O ratio: In men: 0.51 In women: 0.30 | DCA |

(*Continued*)

**Table 1.** (Continued)

| Reference | Risk score & Predictors | Country (WHO region) | Data source & follow-up time | Study Population | Outcome | Sample Size (Outcome) | Modeling method | Model Predictive Performance | | |
|---|---|---|---|---|---|---|---|---|---|---|
| | | | | | | | | Discrimination | Calibration | Clinical usefulness |
| Liu et al.2022 [57] | Age, smoking status, hypertension treatment, SBP, TC, HDL-C | China (Western Pacific Region) | CHERRY study, a general population-based cohort study Follow-up time (median): 4.60 years | Aged 40 to 79 years without prior history of ASCVD at baseline from 2010 to 2016 | **5-year risk of ASCVD** (i.e. nonfatal or fatal stroke, nonfatal MI, and CV death) | 226406 individuals; 105848 men & 120558 women (5362 events) | Pooled cohort equations | C-statistic (95% CI) In men: 0.763 (0.754,0.773) In women: 0.820 (0.812,0.829) | Calibration plot The H-L $\chi^2$: in men 953.23 (P<0.001) In women 341.64 (P<0.001) EO ratio: In men: 1.63 In women: 0.66 Calibration slope: In men: 1.18 In women: 1.08 | --- |
| **RISK SCORE: SCORE risk model [47]** | | | | | | | | | | |
| Sawano et al. 2016 [58] | Age, TC, SBP, smoking status | Japan (Western Pacific Region) | NIPPON DATA80 cohort study Follow-up time (median): 9.83 years | Aged 40 to 64 years, SBP between 100 and 180 mmHg, and TC level < 303 mg/dL | 10-year risk of CV death (i.e. CV death, and coronary death) | 4842 individuals; 2094 men & 2748 women (44 events) | CoxPH | C-statistic (95% CI) In men: 0.71(0.69,0.73) In women 0.71 (0.70, 0.73) | Calibration plot The H-L $\chi^2$: In men 0.749 (P = 0.38) In women: 1.39 (P = 0.24) | --- |
| Fahimfar et al. 2022 [38] | Age, SBP, TC, current smoking | Iran (Eastern Mediterranean Region) | Four population-based cohorts (TLGS, ICS, GCS & ShECS) Follow-up time: > 10 years in TLGS & ICS, and 5 years in ShECS & GCS | Aged 40 to 80 years, Without a history of CVD at baseline | 10-year risk of CV mortality (i.e. fatal IHD, sudden cardiac death or stroke) | 24427 individuals; 11187 men & 13240 women (437 events) | Weibull parametric model | C-statistic (95% CI) In men: 0.784 (0.756,0.812) In women: 0.780 (0.744, 0.815) | Calibration plot Calibration slope(95% CI): In men: 0.84 (0.70, 0.99) In women: 0.62 (0.46, 0.78) EO ratio: In men: 1.02 In women: 0.95 | DCA |
| **RISK SCORE: Globorisk risk model [19]** | | | | | | | | | | |

*(Continued)*

**Table 1.** (Continued)

| | | | | | | CoxPH | C-statistic (95% CI) | Calibration plot | DCA |
|---|---|---|---|---|---|---|---|---|---|
| Fahimfar et al. 2022 [38] | Age, SBP, TC, current smoking, diabetes | Iran (Eastern Mediterranean Region) | Four population-based cohorts (TLGS, ICS, GCS & ShECS) **Follow-up time:** > 10 years in TLGS & ICS, and >5 years ShECS & GCS2 | Aged 40 to 80 years Without a history of CVD at baseline | 10- year risk of CV mortality (i.e. fatal IHD, sudden cardiac death or stroke) | 24427 individuals; 11187 men & 13240 women (437 events) | | In men: 0.793(0.766, 0.820) In women: 0.793 (0.757,0.829) | Calibration slope(95% CI): In men: 1.30 (0.94,1.66) In women: 1.24 (0.76,1.71) EO ratio: In men: 1.02 In women: 0.95 | |

**RISK SCORE: WHO CVD risk charts for East Asia (lab-based)** [48]

| | | | | | | | | | |
|---|---|---|---|---|---|---|---|---|---|
| Li et al. 2021 [42] | Sex, age, current smoking, SBP, diabetes, TC | China (Western Pacific Region) | China-PAR cohort Follow-up time (average): 13.64 years | Aged 40 to 80 years | 10- year risk of CVD (i.e. acute MI, fatal CHD, and nonfatal or fatal stroke) | 93234 individuals; 38537 men & 54697 women (1849 events) | --- | In men: 0.759(0.740, 0.779) In women: 0.752 (0.728, 0.777) | The H-L $\chi$2: In men 321.55 (P<0.001) In women: 280.69 (P<0.001) | --- |

**RISK SCORE: WHO CVD risk charts for East Asia (non-lab based)** [48]

| | | | | | | | | | |
|---|---|---|---|---|---|---|---|---|---|
| Li et al. 2021 [42] | Sex, age, current smoking, SBP, BMI | China (Western Pacific Region) | China-PAR cohort Follow-up time (average): 13.64 years | Aged 40 to 80 years | 10- year risk of CVD (i.e. acute MI, fatal CHD, and nonfatal or fatal stroke) | 93234 individuals; 38537 men & 54697 women (1849 events) | --- | In men: 0.726(0.744, 0.781) In women: 0.754 (0.731,0.777) | The H-L $\chi$2: In men: 386.18 (P<0.001) In women: 439.99 (P<0.001) | --- |

WHO: World Health Organization; TC: Total Cholesterol; HDL-C: High-density lipoprotein cholesterol; BP: Blood pressure; MI: Myocardial Infarction. CHD: Coronary heart disease; CoxPH: Cox Proportional Hazard model; C-statistic: Concordance statistic; H-L $\chi$ 2: Hosmer-Lemeshow Chi squared test. SBP: Systolic blood pressure; DBP: Diastolic blood pressure; TG: Triglycerides; LDL-C: Low density lipoprotein cholesterol; TIA: Transient ischemic attack; PVD: Peripheral vascular disease; IHD: Ischemic Body mass index; CV: Cardiovascular; TLGS: Tehran Lipid and Glucose Study; CVD: Cardiovascular disease; HF: Heart failure; HTN: Hypertension; FPG: Fasting plasma glucose; BMI: heart disease; ASCVD: atherosclerotic cardiovascular disease; UA: Unstable angina; AF: Atrial fibrillation; EO ratio: Expected to observed ratio; PCEs: Pooled cohort equations; ACC/AHA: American College of Cardiology/ American Heart Association; SD: Standard deviation; CI: Confidence interval; CHERRY: CHinese Electronic health Records Research in Yinzhou; DCA: Decision carve analysis; H-L: Hosmer-Lemeshow test; mg/dL: Milligram per deciliter; ICS: Isfahan Cohort Study; GCS: Golestan Cohort Study; ShECS: Shahroud Eye Cohort Study; China-PAR: Prediction for atherosclerotic cardiovascular disease risk in China.

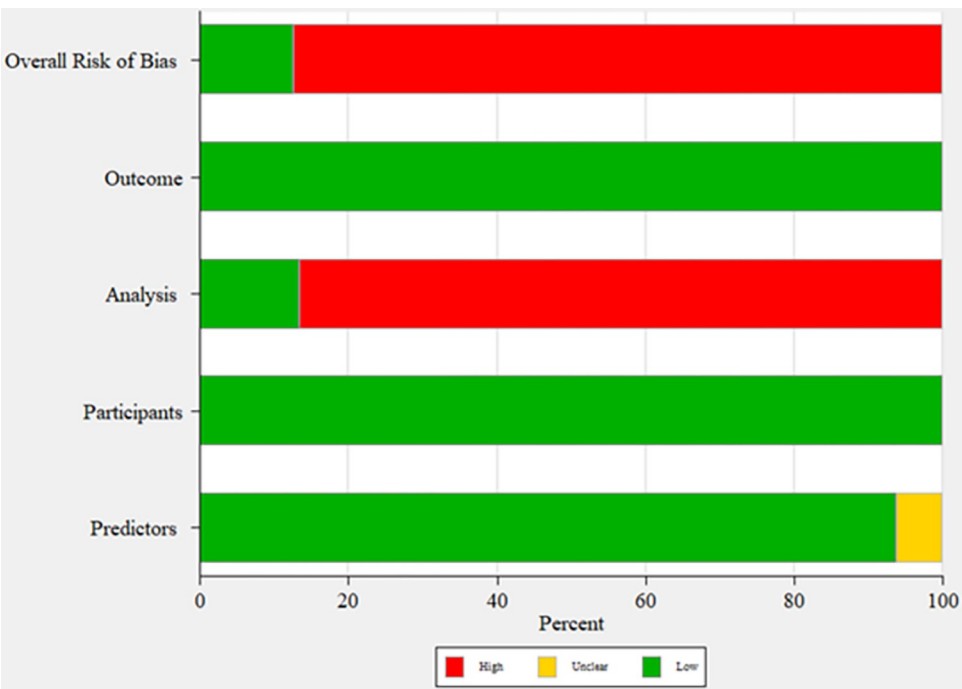

**Fig 2. Summary of risk of bias analysis by PROBAST (prediction model risk of bias assessment tool).**

## Performance of the validated models

The statistical performances, including discriminative ability using concordance statistics and calibration using a calibration plot, were reported in all validated models. The clinical usefulness of decision curve analysis was reported in only six models (29%) compared to the statistical performance reported in all models.

Framingham CHD [44], Framingham general CVD [45], PCEs [46], and SCORE risk models [47] were validated more than once, and we quantitatively synthesized their performance measures. The performance of the Globorisk [19] and WHO [48] models that were validated once is shown in **Table 1**.

## Discrimination

The forest plot in **Fig 3** displays the C-statistic of risk models for both genders. The pooled C-statistic for men ranged from 0.72 (95% CI 0.70 to 0.75 and 95% PI 0.65 to 0.79) for the PCEs to 0.76 (95% CI 0.74 to 0.78 and 95% PI 0.70 to 0.81) for the Framingham general CVD model. In women, it varied between 0.74 (0.95% CI 0.22 to 0.97) for the SCORE and 0.79 (95% CI 0.74 to 0.83 and 95% PI 0.63 to 0.89) for the Framingham general CVD (**Fig 3**).

## Calibration

**Fig 4** provides a Forest plot of the OE ratio by risk models across gender. The Framingham model for CHD indicated overprediction for both men and women. The Framingham general CVD and SCORE models showed an overprediction among men and an underprediction among women. The PCEs indicated an underprediction in both men and women (**Fig 4**).

According to **Fig 4**, the pooled OE ratio for men ranged from 0.21 (95% CI 0.018 to 2.49) for the Framingham CHD model to 1.11 (95% CI 0.65 to 1.89, and 95% PI 0.21 to 5.77) for the

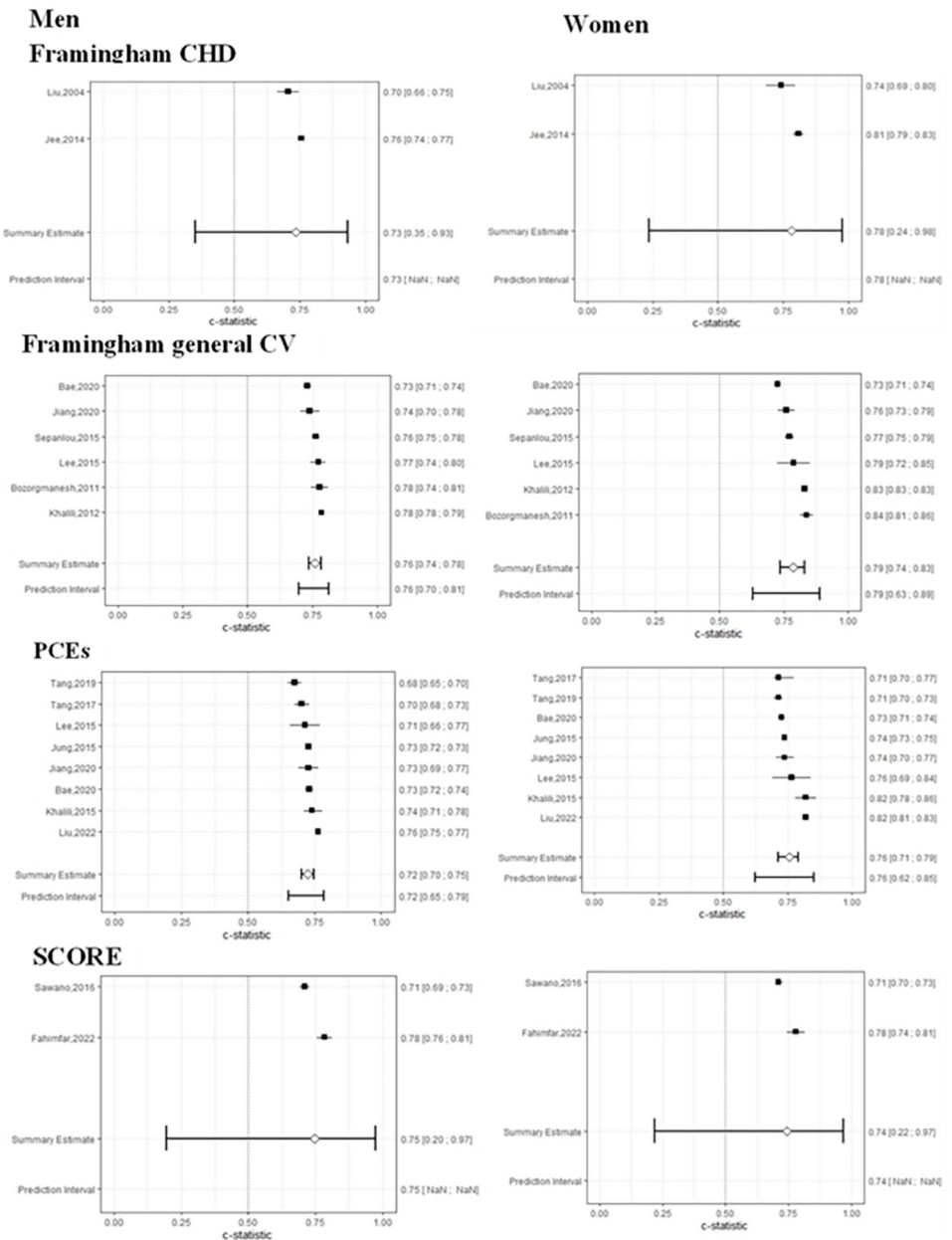

**Fig 3.** Forest plot of C-statistic by risk models among men (left) and women (right).

PCEs. In women, it varied between 0.28 (95% CI 0.33 to 2.33) for the Framingham CHD model and 1.81(95% CI 0.90 to 3.64, and 95% PI 0.21 to 15.74) for the PCEs (**Fig 4**).

## Sensitivity analysis

The discrimination for the Framingham general CVD model and PCEs did not change after excluding studies that provided a 5-year risk of CVDs in both men and women, while their calibration decreased (**S1 and S2 Figs**).

The C-statistic for the Framingham CHD through the Fixed-effect model was 0.75 (95% CI 0.74 to 0.76) in men and 0.80 (95%CI 0.78 to 0.82) in women. For the SCORE risk model, the

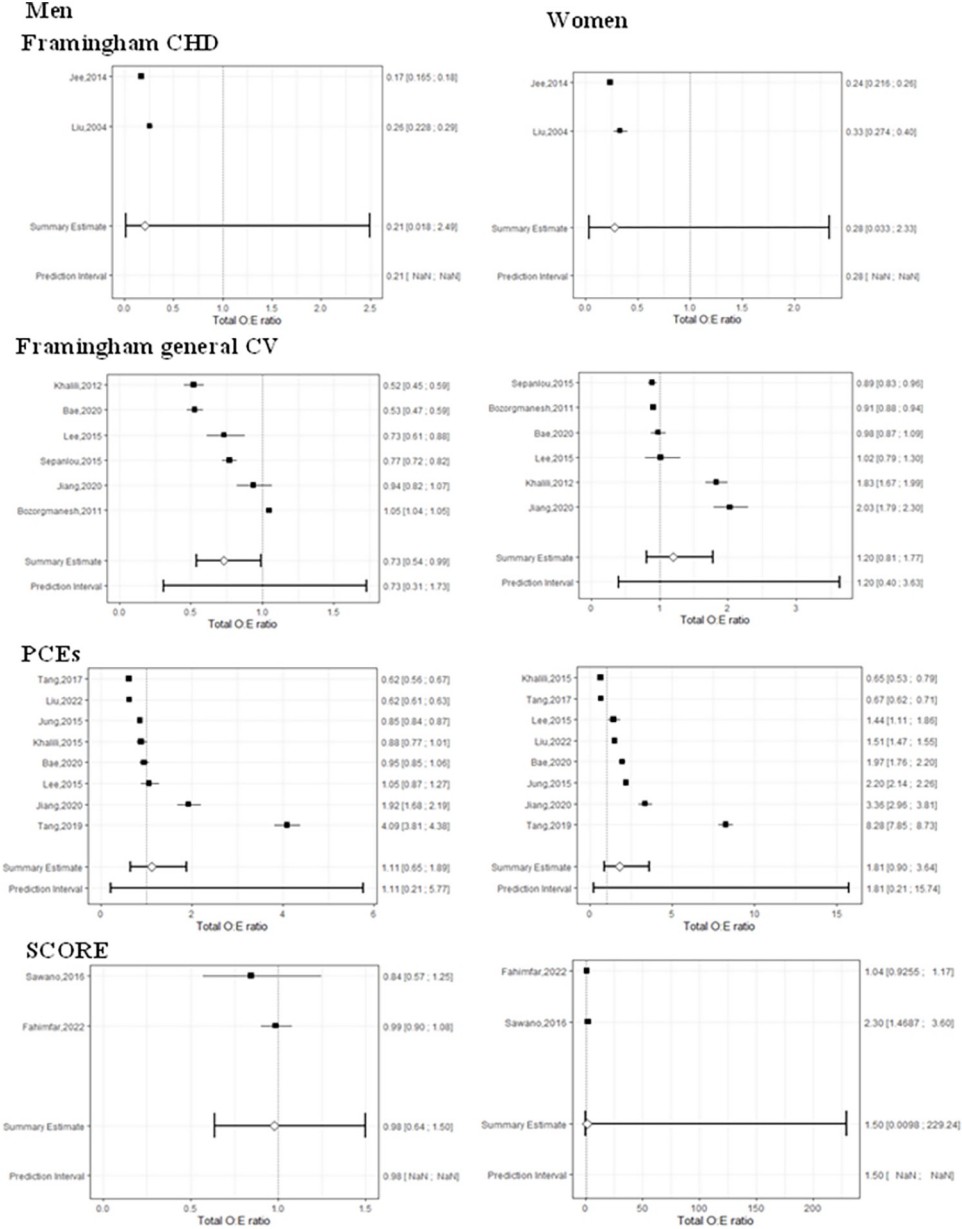

**Fig 4.** Forest plot of OE ratio by risk models among men (left) and women (right).

C-statistics through the Fixed-effect model were 0.73 (95%CI 0.71 to 0.75) and 0.72 (95% CI 0.70 to 0.73) in men and women, respectively.

The OE ratio for the Framingham CHD through the Fixed-effect model was 0.18 (95% CI 0.17 to 19) in men and 0.25 (95%CI 0.23 to 0.28) in women. For the SCORE risk model, the OE ratio was 0.98 (95% CI 0.90 to 1.07) and 1.09 (95% CI 0.98 to 1.22) in men and women, respectively.

## Discussion

To the best of our knowledge, for the first time, we have systematically reviewed the performance of models for predicting the 10-year risk of CVDs in the general population in Asia.

The Framingham CHD, Framingham general CVD, PCEs, SCORE and Globorisk models, and WHO risk charts for East Asia were all risk scores that were externally validated in this region. The Framingham general CVD and PCEs were the most validated tools in this study. Additionally, Globorisk and WHO models were validated once. We found an acceptable discriminative ability for all models in men and women. The Framingham CHD model overestimated the risk in both genders. However, the Framingham general CVD and SCORE model overestimated the risk in men and underestimated it in women. The PCEs underestimated the risk in both men and women.

A systematic review and meta-analysis by Damen et al. in 2019 [21] evaluated the performance of the Framingham CHD model and PCEs. In 2022, Zhiting et al. [24] assessed the performance of the Framingham general CVD model and PCEs in the general Chinese population through a systematic review. Both systematic reviews focused on the most validated models. Neither of the two systematic reviews considered the SCORE model, which was noticed in the current review. Therefore, they did not provide a complete picture of the validated model's performance in Asia, which necessitated this study.

Discrimination and calibration are model performance evaluation's two most important aspects [59]. In a systematic review, 64% of models had the C-statistic or area under the receiver operating characteristic curve (AUC-ROC) as a measure of discrimination. Calibration plot (26%), observed to the expected ratio (26%), or Hosmer-Lemeshow test (14%) were among the methods used to report calibration [22]. It was shown that reporting discrimination increased from 14% in 1976 to 77% in 2013. In contrast, reporting calibration decreased from 76% in 1976 to 59% in 2013 [22]. In this review, all studies reported a C-statistic to demonstrate the discriminative ability of prediction models, and they used calibration plots to show the agreement between the observed and expected event probabilities. In line with the previous review [22], some studies published before 2015 did not report additional calibration measures, such as the OE ratio. Calibration is often considered the Achilles heel of prediction analysis, as it enables us to understand the model performance in a particular setting, identify where predictions may be inaccurate, and determine if the model needs to be updated [60]. Discrimination and calibration are essential but not sufficient for clinical usefulness. Reporting decision curves or net benefit is necessary to gain further insight into the feasibility of such models for clinical use [61]. In this review, only one-third of the validated models provided this information. As suggested by the TRIPOD guideline [62], it is advisable to present performance measures such as discrimination, calibration, and decision curve or net benefit when using a model to assist clinical decision-making [61].

In this review, the discriminative ability of the models varied between 0.7 and 0.8. In the previous review by Damen et al. in 2019, the discriminative ability of the Framingham CHD model and PCEs ranged between 0.68 and 0.74 [23]. It was also between 0.72 and 0.76 for the Framingham general CVD model and PCEs in the Chinese population [24]. The absence of a consistent metric in models and significant differences in case mix may affect the accuracy of this method. For instance, where the case mix is too homogeneous, we found poor discrimination. As a result, it is recommended to consider case-mix variation when evaluating discrimination measures [63].

A systematic review found that the Framingham CHD model and PCEs overestimated the risk of cardiovascular disease, particularly in higher-risk populations [23]. In a systematic review of the general Chinese population, the Framingham general CVD model and PCEs overestimated the risk in men and underestimated it in women, respectively [24]. In our study, the Framingham CHD model overestimated the risk in line with previous studies. However, the Framingham general CVD model and PCEs behaved differently. SCORE model behaved like the Framingham general CVD model. The difference in model calibration could

be attributed to case-mix [23,64] and the healthcare system difference [23]. Using a model based on a particular region is inaccurate when assessing the actual risk in another area [24]. Applying these models as initially developed to make treatment decisions could result in either over- or undertreatment of individuals, leading to unnecessary burden to individuals and the health care system. As a result, it is recommended that none of these models be put to use without recalibrating the baseline risk or hazard according to the local setting [23].

There were only two studies for the Framingham CHD and SCORE risk models, and we combined the performance measures through the random-effects model. Validation studies are different in design, conduct, and case mix; we cannot attribute between study variations to chance only. Therefore, it is necessary to consider the heterogeneity through the meta-analysis rather than ignore it [28], and using random-effects models is generally recommended [34]. On the other hand, despite the technical possibility of applying random-effects meta-analysis, it is not sensible in the case of two studies. Thus, the Fixed-effect model is recommended unless there are strong arguments against the common effect assumption [35]. Therefore, we also performed a fixed-effect meta-analysis for the aforementioned models. Random-effects models produce wider 95%CI compared to the Fixed-effect model, which indicates lower certainty on the pooled estimates and the need for more validation studies in this region.

## Study limitations

Our study had several limitations. First, our source of information was restricted to what the authors provided in the primary studies, and we had to exclude one study with insufficient data, which we could not obtain from the authors. Also, we were unable to perform a meta-analysis for the WHO risk charts and Globorisk model because the full text was not available despite contact with the authors, and one-time validation, respectively. On the other hand, WHO risk charts and Globorisk models have been externally validated in their original papers. However, we did not consider the external validation results as part of model development in this paper. In addition, we had a small number of studies and used a random effects model, in which the point estimate and confidence interval may provide a false sense of certainty. However, few suitable alternatives exist when limited studies are available [65]. Second, for some studies before 2015 that did not report the number of observed and expected events, we had to use the calibration plot to estimate the OE ratio without the authors'proof. Third, due to the limited number of validated studies [28], we did not perform a meta-regression analysis to detect possible reasons for heterogeneity for each model. Fourth, we compared the statistical performance measures, but we were unable to combine the models'clinical usefulness to find the therapeutic threshold for each risk model because there were only a few studies. Fifth, over half of the studies were of low quality because they did not handle missing data well, which might affect the study measures. Sixth, because of the descriptive nature of the study and due to the limited number of included studies for each model (range of 2 to 8 studies), we did not evaluate the publication bias.

## Conclusion

In this study, the Framingham CHD and general CVD, PCEs, and SCORE models performed differently to predict the 10-year risk of CVDs in Asia. Although these four models indicated an acceptable level of discrimination in this region, their calibration differed among men and women. The Framingham CHD model and PCEs over-and underestimated the risk, respectively. At the same time, the Framingham general CVD and SCORE models overestimated the risk in men and underestimated it in women. If a local recalibration and updating are not made, it will lead to over- or undertreatment in clinical practice and additional harm and cost.

Further research should focus on external validation of the existing prediction models or recalibration and updating these models for local settings to guide CVD prevention in this region and target the population based on risk predictions. In addition, these models' decision curves and net benefits should be assessed to help clinical decision-making.

## Supporting information

**S1 Checklist. PRISMA 2020 checklist.**
(PDF)

**S1 Fig. Forest plot of C-statistic by risk models among men (left) and women (right) after excluding studies provided 5-year risk of CVDs.**
(TIF)

**S2 Fig. Forest plot of OE ratio by risk models among men (left) and women (right) after excluding studies provided 5-year risk of CVDs.**
(TIF)

**S1 Table. Search strategy used to retrieve related documents (Search date: November 28 2022).**
(DOCX)

**S2 Table. Summary of risk of bias analysis by PROBAST for included studies.**
(DOCX)

## Acknowledgments

This article relates to the dissertation of Mahin Nomali for a Ph.D. degree in Epidemiology (no. 240/1071) at the School of Public Health of Tehran University of Medical Sciences (TUMS). We would like to thank the University of Santiago de Compostela for facilitating a year of access to the databases through the USC student service account.

## Author Contributions

**Conceptualization:** Mahin Nomali, Davood Khalili, Mehdi Yaseri, Aryan Ayati, Hossein Navid, Saharnaz Nedjat.

**Data curation:** Mahin Nomali, Mehdi Yaseri, Saharnaz Nedjat.

**Formal analysis:** Mahin Nomali, Hossein Navid.

**Investigation:** Mahin Nomali.

**Methodology:** Mahin Nomali, Davood Khalili, Aryan Ayati.

**Supervision:** Mohammad Ali Mansournia, Hossein Navid, Saharnaz Nedjat.

**Validation:** Davood Khalili, Mehdi Yaseri, Mohammad Ali Mansournia, Saharnaz Nedjat.

**Writing – original draft:** Mahin Nomali.

**Writing – review & editing:** Aryan Ayati, Saharnaz Nedjat.

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
