## [Decision Letter · Decision Letter 0]

17 Aug 2023

PONE-D-23-20596Perfromance of the models predicting 10- year risk of cardiovascular diseases in Asia: A systematic review and prediction model meta-analysisPLOS ONE

Dear Dr. Nedjat,

Thank you for submitting your manuscript to PLOS ONE. After careful consideration, we feel that it has merit but does not fully meet PLOS ONE’s publication criteria as it currently stands. Therefore, we invite you to submit a revised version of the manuscript that addresses the points raised during the review process.

We look forward to receiving your revised manuscript.

Kind regards,

Hean Teik Ong

Academic Editor

PLOS ONE

5. We notice that your supplementary figure and table are included in the manuscript file. Please remove them and upload them with the file type 'Supporting Information'. Please ensure that each Supporting Information file has a legend listed in the manuscript after the references list.

Additional Editor Comments:

PLEASE ADDRESS COMMENTS OF REVIEWERS AND MAKE MAJOR REVISIONS TO ARTICLE.

Reviewers' comments:

Reviewer's Responses to Questions

**Comments to the Author**

1. Is the manuscript technically sound, and do the data support the conclusions?

Reviewer #1: Yes

Reviewer #2: Yes

2. Has the statistical analysis been performed appropriately and rigorously? 

Reviewer #1: Yes

Reviewer #2: Yes

3. Have the authors made all data underlying the findings in their manuscript fully available?

Reviewer #1: Yes

Reviewer #2: Yes

4. Is the manuscript presented in an intelligible fashion and written in standard English?

Reviewer #1: Yes

Reviewer #2: Yes

5. Review Comments to the Author

Reviewer #1: I would like to thank the authors to do this valuable meta-analysis. Prediction model meta-analysis is one of the advanced methods, which their related guidelines were released in 2018for the first time. Thus, it is a practical paper and helps researchers, policy makers, andclinicians to be aware of the models` performance in their own region. On the other hand, dueto rising prevalence of CVDs in Asia and lack of risk assessment tools in this region, andbecause of recommendation of AHA and ESC guidelines to include individual riskassessment in the primary prevention program, the accuracy of the risk estimates by the riskassessment tools is crucial. I reviewed the manuscript. It was written well. However, there aresome comments as follows: Abstract- It is recommended to write this section according to the PRISMA 2020 for abstractchecklist to improve the quality of reporting. I know that report all the information inthe manuscript. - In this paragraph, “Out of 1315 initial records, 16 studies were included, with 21external validations of six models in Asia. The pooled concordance statistics ofFramingham coronary heart disease (CHD) and Framingham general CVD model,pooled cohort equations (PCEs), and SCORE model were 0.73, 0.76, 0.72 and 0.75 inmen, and 0.78, 0.79, 0.76 and 0.74 in women, respectively. The pooled OE ratio forthe four afformantined models were 0.21, 0.73, 1.11 and 0.98 in men, and 0.28, 1.20,1.81 and 1.50 in women, respectively.”, just point estimates have been reported. Iknow that you reported the 95% CI in the full text. It is recommended to report 95%CI, as well. OR if you did not report the 95%CI because of word count limitation, youcan provide just range of the C-index for included models in the meta-analysis. - According to PRISMA 2020 for abstract checklist, please name the databases. Youjust mentioned “Various databases”. - “The risk of bias was assessed using a specific tool”, which tool? Please mention it. - “Meta-analyses were performed using statistical measures for discrimination andcalibration, nclusing concordance statistic (C-statistics) and observed to expected(OE) ratio, respecetively”. Please specify the methods used to present and synthesisresults. - Please check the whole manuscript in terms of spelling and grammar (i.e. nclusing,respecetively,… )

Introduction - In this paragraph, “Cardiovascular diseases (CVDs) are the leading cause ofworldwide disease burden. The number of total CVD cases has nearly doubled from271 million in 1990 to 523 million in 2019, and the number of CVD deaths increasedfrom 12.1 million in 1990 to 18.6 million in 2019 [1]. In Asia, between 1990 and2019, CVD deaths surged from 5.6 million to 10.8 million [2].”, you providedinformation about global burden of disease. Please provide much information aboutAsia, as well. Methods: - The authors explained the Statistical analysis well. However, there is no informationabout publication bias. I know that PMMA are different from conventional MA andthey have their own guidelines. I also checked the related meta-analyses you cited inthe introduction section and they did not provide the funnel plot, etc as well. If it ispossible, please provide it or mention it in the study limitation. Results: - In Fig.2, the results of risk of bias assessment using PROBAST were reported totally.Please provide the total quality score in the Table 1, as well. - In Table 2, the predictors of the original models and validated models are similar.Thus, it can be omitted.

Reviewer #2: The manuscript is structured in a clear and organized manner. The manuscript is a systematic review and meta-analysis of existing prediction models for cardiovascular diseases (CVDs) in the general population in Asia. The review includes studies that met specific inclusion criteria and used statistical measures to assess the performance of the prediction models. The file discusses the increasing prevalence of CVDs in Asia and the behavioral, environmental, and social factors contributing to this trend. The authors interpret the results of the study in the discussion section, highlighting the limitations of current prediction models and the need for further research to develop more accurate models that account for the unique risk factors in the Asian population. The findings of the study can inform clinical practice and public health policy in Asia by providing insights into the validity of existing prediction models for CVDs. I think it is a valuable article and could be considered to be published.

6. PLOS authors have the option to publish the peer review history of their article (what does this mean?). If published, this will include your full peer review and any attached files.

Reviewer #1: **Yes: **Meisam Akhlaghdoust

Reviewer #2: No

---

## [Author Response · Author response to Decision Letter 0]

30 Aug 2023

Manuscript title: Validity of the models predicting 10- year risk of cardiovascular diseases in Asia: A systematic review and prediction model meta-analysis

Dear Editorial team of PLOSONE journal,

On behalf of all co-authors, I would like to thank you and the reviewers for the accurate review and valuable comments on our work that helped us improve our manuscript's quality considerably. We appreciate all the efforts and time put into this. We carefully addressed the insightful comments of the reviewers. The following pages include a point-by-point response to the comments with references to the changes made in the manuscript. As instructed, we have submitted a revised version of the manuscript with the changes tracked in the manuscript as well as a clean version.

We are looking forward to receive your kind response and decision.

Yours sincerely,

Saharnaz Nedjat 

Professor of Epidemiology (MD, Ph. D)

Address: Keshavarz Blvd., Poursina Street, Department of Epidemiology and Biostatistics, School of Public Health, Tehran University of Medical Sciences, Tehran, Iran

Postal code: 1417613151

Tel: 982188992969

Fax: 982188989123

Email: nejatsan@tums.ac.ir, drsaharnaznedjat@gmail.com

ORCID: 0000-0002-0966-727X 

Response: We have revised the manuscript accordingly. 

Response: We have added the Excel file of our review data to the submission. We have also added DOIs in the references section.

3. PLOS requires an ORCID iD for the corresponding author in Editorial Manager on papers submitted after December 6, 2016. Please ensure that you have an ORCID iD and that it is validated in Editorial Manager. To do this, go to 'Update my Information' (in the upper left-hand corner of the main menu), and click on the Fetch/Validate link next to the ORCID field. This will take you to the ORCID site and allow you to create a new iD or authenticate a pre-existing iD in Editorial Manager. Please see the following video for instructions on linking an ORCID iD to your Editorial Manager account: https://www.youtube.com/watch?v=_xcclfuvtxQ

Response: We have added the ORCID of the corresponding author to the manuscript and the submission system (0000-0002-0966-727X)

Response: Thank you for reminding this important point. We removed the "ethical approval" 

section, and we mentioned, "This was a systematic review and prediction model meta-analysis, which was approved by the Research Ethics Committee (REC) of School of Public Health & Allied Medical Sciences at TUMS on March 12, 2022 (approval ID: IR.TUMS.SPH.REC.1400.353)" at the first paragraph of the methods section based on you recommended. 

5. We notice that your supplementary figure and table are included in the manuscript file. Please remove them and upload them with the file type 'Supporting Information'. Please ensure that each Supporting Information file has a legend listed in the manuscript after the references list.

Response: We have removed the supplementary materials from the manuscript and moved it to a separate file.

5. Review Comments to the Author

Reviewer #1: 

I would like to thank the authors to do this valuable meta-analysis. Prediction model meta-analysis is one of the advanced methods, which their related guidelines were released in 2018 for the first time. Thus, it is a practical paper and helps researchers, policy makers, and clinicians to be aware of the models` performance in their own region. On the other hand, due to rising prevalence of CVDs in Asia and lack of risk assessment tools in this region, and because of recommendation of AHA and ESC guidelines to include individual risk assessment in the primary prevention program, the accuracy of the risk estimates by the risk assessment tools is crucial. I reviewed the manuscript. It was written well. However, there are some comments as follows: 

Abstract:

- It is recommended to write this section according to the PRISMA 2020 for abstract checklist to improve the quality of reporting. I know that report all the information in the manuscript.

Response: Thank you very much. As you recommended, the abstract section was written according to the PRISMA 2020 for abstract. Thus, removed the background and limited this section to the review objective. We also limited the methods section to eligibility criteria, information sources, risk of bias, and synthesis of results. 

- In this paragraph, "Out of 1315 initial records, 16 studies were included, with 21external validations of six models in Asia. The pooled concordance statistics of Framingham coronary heart disease (CHD) and Framingham general CVD model, pooled cohort equations (PCEs), and SCORE model were 0.73, 0.76, 0.72 and 0.75 in men, and 0.78, 0.79, 0.76 and 0.74 in women, respectively. The pooled OE ratio for the four aforementioned models were 0.21, 0.73, 1.11 and 0.98 in men, and 0.28, 1.20,1.81 and 1.50 in women, respectively.", just point estimates have been reported. I know that you reported the 95% CI in the full text. It is recommended to report 95%CI, as well. OR if you did not report the 95%CI because of word count limitation, you can provide just range of the C-index for included models in the meta-analysis.

Response: Thank you so much for this comment. Point estimate and 95% CI is very important. Thus, we revised the results "The validated models consisted of Framingham models, pooled cohort equations (PCEs), SCORE, Globorisk, and WHO models, which the results of the first four models were combined. The pooled C-statistic for men ranged from 0.72 (95% CI 0.70 to 0.75; PCEs) to 0.76 (95% CI 0.74 to 0.78; Framingham general CVD). In women, it varied from 0.74 (95% CI 0.22 to 0.97; SCORE) to 0.79 (95% CI 0.74 to 0.83; Framingham general CVD). The pooled OE ratio for men ranged from 0.21 (95% CI 0.018 to 2.49; Framingham CHD) to 1.11 (95% CI 0.65 to 1.89; PCEs). In women, it varied from 0.28 (95% CI 0.33 to 2.33; Framingham CHD) to 1.81 (95% CI 0.90 to 3.64; PCEs)." 

- According to PRISMA 2020 for abstract checklist, please name the databases. You just mentioned "Various databases". 

Response: As you recommended, we revised "Various databases, including PubMed, Web of Science conference proceedings citation index, Scopus, Global Index Medicus of the World Health Organization (WHO), and Open Access Thesis and Dissertations (OATD) were searched up to November 2022" 

- "The risk of bias was assessed using a specific tool", which tool? Please mention it. 

Response: Based on your valuable comments, we revised to "The risk of bias was assessed using the PROBAST, prediction model risk of bias assessment tool." 

- "Meta-analyses were performed using statistical measures for discrimination and calibration, including concordance statistic (C-statistics) and observed to expected (OE) ratio, respectively". Please specify the methods used to present and synthesis results. 

Response: Many thanks for this comment. Therefore, we revised this section "Meta-analyses were performed using the random effects model, focusing on the C-statistic as a discrimination index and the observed to the expected ratio (OE) as a calibration index." 

- Please check the whole manuscript in terms of spelling and grammar (i.e. nclusing, respecetively,…)

Response: A comprehensive spelling and grammar assessment was performed for the manuscript, thank you.

Introduction: 

 - In this paragraph, "Cardiovascular diseases (CVDs) are the leading cause of worldwide disease burden. The number of total CVD cases has nearly doubled from271 million in 1990 to 523 million in 2019, and the number of CVD deaths increased from 12.1 million in 1990 to 18.6 million in 2019 [1]. In Asia, between 1990 and2019, CVD deaths surged from 5.6 million to 10.8 million [2].", you provided information about global burden of disease. Please provide much information about Asia, as well. 

Response: Yes, that's right. Thank you. We reviewed the related papers and revise the introduction as "In Asia, CVD is the major cause of death [2], in which CVD deaths surged from 5.6 million to 10.8 million between 1990 and 2019 [3]. 

It is expected that the number of people affected by CVD will significantly rise due to factors such as population growth and aging, particularly in Northern Africa and Western Asia, Central and Southern Asia, Latin America, and the Caribbean, and Eastern and Southeastern Asia, where the percentage of elderly individuals is anticipated to double by 2050 [1]. On the other hand, Since Asian countries have become more Westernized, the have been eating more fat, which has led to an increase in serum total cholesterol, and may have contributed to an increase in coronary heart disease in this region [4]." 

Methods: 

- The authors explained the Statistical analysis well. However, there is no information about publication bias. I know that PMMA are different from conventional MA and they have their own guidelines. I also checked the related meta-analyses you cited in the introduction section and they did not provide the funnel plot, etc as well. If it is possible, please provide it or mention it in the study limitation. 

Response: Thank you so much for this point. Publication bias is the selective publication of research studies based on their results. Therefore, studies with positive findings are more likely to be published than studies with negative findings. In this SR, we did not have studies with positive or negative results, and we did not evaluate any relationship in our study. Thus, we did not assess the publication bias because of the descriptive nature of this MA. On the other hand, we had less than 10 studies for each included model. Thus, we could not evaluate the publication bias. 

However, due to your valuable comment, we added, "Sixth, because of the descriptive nature of the study and due to the limited number of included studies for each model (range of 2 to 8 studies), we did not evaluate the publication bias." to our study limitations. 

Results: 

- In Fig.2, the results of risk of bias assessment using PROBAST were reported totally. Please provide the total quality score in Table 1, as well. 

Response: We provided the ROB assessment results as a separate table in the Supplementary. In the risk of bias assessment of methods section, we also mentioned that "Also, the detailed information has been provided as Table S2". 

- In Table 2, the predictors of the original models and validated models are similar. Thus, it can be omitted.

Response: Thank you. But, it is better to be kept on the table. Because in the PMMA, both predictors and outcomes are very important. 

Reviewer #2: 

The manuscript is structured in a clear and organized manner. The manuscript is a systematic review and meta-analysis of existing prediction models for cardiovascular diseases (CVDs) in the general population in Asia. The review includes studies that met specific inclusion criteria and used statistical measures to assess the performance of the prediction models. The file discusses the increasing prevalence of CVDs in Asia and the behavioral, environmental, and social factors contributing to this trend. The authors interpret the results of the study in the discussion section, highlighting the limitations of current prediction models and the need for further research to develop more accurate models that account for the unique risk factors in the Asian population. The findings of the study can inform clinical practice and public health policy in Asia by providing insights into the validity of existing prediction models for CVDs. I think it is a valuable article and could be considered to be published.

Response: Thank you very much for your positive perception of this article. 

---

## [Decision Letter · Decision Letter 1]

20 Sep 2023

Validity of the models predicting 10- year risk of cardiovascular diseases in Asia: A systematic review and prediction model meta-analysisDear Dr. Nedjat,We are pleased to inform you that your manuscript has been judged scientifically suitable for publication and will be formally accepted for publication once it meets all outstanding technical requirements.Within one week, you will receive an e-mail detailing the required amendments. When these have been addressed, you will receive a formal acceptance letter and your manuscript will be scheduled for publication.An invoice for payment will follow shortly after the formal acceptance. To ensure an efficient process, please log into Editorial Manager at http://www.editorialmanager.com/pone/, click the Update My Information link at the top of the page, and double check that your user information is up-to-date. If you have any billing related questions, please contact our Author Billing department directly at authorbilling@plos.org.If your institution or institutions have a press office, please notify them about your upcoming paper to help maximize its impact. If they’ll be preparing press materials, please inform our press team as soon as possible -- no later than 48 hours after receiving the formal acceptance. Your manuscript will remain under strict press embargo until 2 pm Eastern Time on the date of publication.

---

## [Editor Report · Acceptance letter]

25 Sep 2023

PONE-D-23-20596R1 

Validity of the models predicting 10- year risk of cardiovascular diseases in Asia: A systematic review and prediction model meta-analysis 

Dear Dr. Nedjat:

I'm pleased to inform you that your manuscript has been deemed suitable for publication in PLOS ONE. Congratulations! Your manuscript is now with our production department. 

Kind regards, 

on behalf of

Dr. Hean Teik Ong 

Academic Editor

PLOS ONE